# Physicochemical, Complexation and Catalytic Properties of Polyampholyte Cryogels

**DOI:** 10.3390/gels5010008

**Published:** 2019-02-21

**Authors:** Sarkyt E. Kudaibergenov

**Affiliations:** 1Institute of Polymer Materials and Technology, Microregion “Atyrau 1”, house 3/1, Almaty 050019, Kazakhstan; skudai@mail.ru; Tel.: +7-727-337-70-47; 2Laboratory of Engineering Profile, K.I. Satpayev Kazakh National Research Technical University, Satpayev Str. 22, Almaty 050013, Kazakhstan

**Keywords:** polyampholyte cryogels, swelling-deswelling, complexation, protein imprinting, immobilization of metal nanoparticles, catalysis, hydrogenation, oxidation, flow-through catalytic reactor

## Abstract

Polyampholyte cryogels are a less considered subject in comparison with cryogels based on nonionic, anionic and cationic precursors. This review is devoted to physicochemical behavior, complexation ability and catalytic properties of cryogels based on amphoteric macromolecules. Polyampholyte cryogels are able to exhibit the stimuli-responsive behavior and change the structure and morphology in response to temperature, pH of the medium, ionic strength and water–organic solvents. Moreover, they can uptake transition metal ions, anionic and cationic dyes, ionic surfactants, polyelectrolytes, proteins, and enzymes through formation of coordination bonds, hydrogen bonds, and electrostatic forces. The catalytic properties of polyampholyte cryogels themselves and with immobilized metal nanoparticles suspended are outlined following hydrolysis, transesterification, hydrogenation and oxidation reactions of various substrates. Application of polyampholyte cryogels as a protein-imprinted matrix for separation and purification of biomacromolecules and for sustained release of proteins is demonstrated. Comparative analysis of the behavior of polyampholyte cryogels with nonionic, anionic and cationic precursors is given together with concluding remarks.

## 1. Introduction

The basic terms and definitions [1] together with the history of polymeric cryogels [2,3] including the synthesis, structure-property relationships, and properties [4,5] are comprehensively reviewed and systematized in the fundamental works of V.I. Lozinsky and O. Okay [6]. Preparation protocol, as well as the structural and morphological characteristics of cryogels and their application aspects with regard to biotechnology and biomedicine can be found [7,8,9]. Among the polymeric cryogels, the most attention has been paid to nonionic, anionic and cationic cryogels [1,2,3,4,5,6,7,8,9]. Amphoteric nano-, micro- and bulk gels have been the subject of much research [10,11,12,13,14,15,16,17]. Cryogels made of natural polyampholytes—proteins—also belong to amphoteric cryogels. However, in this review we will consider mostly macroporous polyampholyte cryogels of synthetic origin, and mainly focus on recent advances in polyampholyte cryogels prepared from acid–base or anionic–cationic monomer pairs or else betainic monomers in cryoconditions, as listed in Table 1. First, the structural and morphological properties of polyampholyte cryogels in response to pH, temperature, ionic strength and water–organic solvents are discussed. Then the complexation of polyampholyte cryogels with transition metal ions, dyes, surfactants, polyelectrolytes and proteins is outlined. Finally, the design of effective catalytic systems with the help of polyampholyte cryogels within which metal nanoparticles are immobilized will be demonstrated. In the conclusion, the prospects and open problems related to polyampholyte cryogels is considered. The physicochemical, complexing and catalytic properties of conventional polyampholyte hydrogels are considered only when they are compared with the behavior of polyampholyte cryogels.

## 2. Physicochemical Properties of Polyampholyte Cryogels

As seen in Table 1, polyampholyte cryogels can be comprised of primary, secondary and tertiary amine, or quaternary ammonium groups on the one hand, and carboxylic, sulfonic acid or carboxylate, sulfonate moieties on the other. Polyampholyte cryogels, analogous to linear polyampholytes [28], can be defined as pH-dependent “annealed” polyampholyte cryogels in contrast to “quenched” polympholyte cryogels which consist of fully charged monomeric units. When cryogels contain an identical number of pH-dependent acid–base or pH-independent anionic–cationic groups in the same monomer units, they can be referred to as “betainic” or “zwitterionic” cryogels.

The key characteristics of any polyampholytes are the ionization constants of acid–base groups (pK_a_ or pK_b_) and the values of the isoelectric point (IEP) where the whole macromolecule exhibits the quasi-neutral character [28]. Table 2 represents the electrochemical characteristics of allylamine-*co*-acrylic acid-*co*-acrylamide P(AA-*co*-AAc-*co*-AAm) cryogels crosslinked by N,N-methylenebisacrylamide (MBAA) [19,29]. Amphoteric cryogels differ from each other by the content of acid–base and acrylamide (AAm) monomers and are abbreviated as ACG-550, ACG-442, ACG-334, ACG-226 and ACG-118 (Figure 1).

Both the water flux and swelling rate of the ACG cryogels decreased in the order of ACG-118 > ACG-226 > ACG-334 > ACG-442 > ACG-550 (Figure 2 and Figure 3). This sequence is in good agreement with the gradual transition of low-charge-density cryogel (ACG-118) to high-charge-density cryogel (ACG-550), accompanied by decreasing the pore size of cryogels and hydration of ionic groups by water molecules. 

The pH_IEP_ of ACG cryogels were between pH 3.5 and 4.3 (Table 2). Time- and pH-dependent swelling of polyampholyte cryogels is demonstarated in Figure 3 and Figure 4.

Swelling of polyampholyte cryogels of ACG series is fast due to the macroporous structure. Increasing of the acid–base content leads to progressive decreasing of the degree of swelling, water flux and pore size, as a result of formation of strong ionic contacts between the oppositely charged acid–base groups of cryogels. 

As seen in Figure 4, the P(DMAEM-co-AMPS) cryogels are in swollen and collapsed states in acidic (pH 2.1) and alkaline (pH 8.0) conditions, respectively. In a narrow interval of pH between 7.7 and 8.0, P(DMAEM-co-AMPS) cryogels exhibit a pH-sensitive phase transition. The values of *n* found with the equation *kt^n^* = *M*_t_/*M*_∞_, (where *k* is the swelling rate constant, *n* is a characteristic exponent describing the mode of penetrant–water–transport mechanism, *t* is absorption time, *M*_t_ is the mass of water absorbed at time *t*, and *M*_∞_ is the mass of water absorbed at infinite time *t*_∞_) at pH 2.1 are in the range of 0.14–0.30 based on AMPS content indicating the non-Fickian type of water diffusion into the cryogel volume.

Figure 5 demonstrates the values of the pH_IEP_ of the ACG series that are arranged between pH 3.5 and 4.3 [29].

The swelling behavior of polyampholyte cryogels was examined in dependence of the ionic strength adjusted by addition of low-molecular-weight salts [24,29]. In acidic and basic media, the swelling degree of P(AA-*co*-AAc-*co*-AAm) cryogels decreases with growth in ionic strength; in contrast, at the IEP the swelling degree of P(AA-*co*-AAc-*co*-AAm) increases as the ionic strength rises (Figure 6). 

The gradual decrease of the swelling degree of P(AA-*co*-AAc-*co*-AAm) with an increase in ionic strength, is explained by the screening of similarly charged macroions by low-molecular-weight salts, whereas shielding of the electrostatic attraction between the opposite charges in P(AA-*co*-AAc-*co*-AAm) results in swelling. 

The swelling behavior of P(DMAEM-*co*-AMPS) cryogels in aqueous solutions of NaCl, KCl, MgCl_2_, K_2_SO_4_, Na_2_SO_4_, and MgSO_4_ is demonstrated in Figure 7 and Figure 8 [24]. For salts with a common anion (Cl^−^, SO_4_^2−^) and mono- or divalent cations, the swelling decreases in order of K^+^ > Na^+^ > Mg^2+^, while for salts with a common cations (K^+^, Na^+^ and Mg^2+^) but different anions, the swelling decreases in order of SO_4_^2−^ > Cl^−^.

The behavior of polyampholyte cryogels in water–organic solvent mixture has not been evaluated thoroughly. However, their collapsing in water–acetone mixture is supposed to behave as that of polyampholyte hydrogels [16,30]. As seen from scanning electron microscope (SEM) pictures, the average pore size of polybetainic gel in pure water is 30–40 µm (Figure 9). In water–acetone mixture the average pore size decreases down to 15–20 µm. In pure acetone, the pores are fully closed, demonstrating the collapse of the gel structure.

The stress-strain curves of P(DMAEM-*co*-MAA) cryogels are presented in Figure 10 [31]. Increasing the concentration of MBAA strengthens the mechanical properties of cryogels.

Thus, the swelling-deswelling properties of polyampholyte cryogels with respect to pH, ionic strength, the nature of anions and cations and in the mixture of water–organic solvent do not deviate from the common peculiarities of linear and crosslinked amphoteric macromolecules [28]. 

## 3. Complexation of Polyampholyte Cryogels with Transition Metal Ions, Dyes, Surfactants, and Proteins

Due to the presence of acid–base groups, polyampholyte cryogels are able to bind both low- and high-molecular-weight substances. Complexation of polyampholyte cryogels with transition metal ions is accompanied by colorization of samples due to formation of coordination and ionic bonds between metal ions and amine and/or carboxylic groups of cryogels (Figure 11). 

As seen in Table 3, the ACG-334 adsorbs up to 99.9% of metal ions but release only 51–67% of metal ions. This is probably due to the capturing of metal ions in dead volumes of networks that are inaccessible for desoprtion. The high adsorption capacity of polyampholyte and polyelectrolyte cryogels can be used for removal of hazardous metal ions [32,33] and dyes [34,35] from waste water. 

The tertiary amine groups of P(DMAEM-*co*-MAA) cryogel bind the sulfonate groups of methyl orange (MO) and sodium dodecylbenzenesulfonate (SDBS) whereas the carboxylic groups interact with amine groups of methylene blue (MB) and quaternary ammonium groups of cetyltrimethylammonium chloride (CTMAC) [31]. 

Immobilization of CTMAC and SDBS within P(DMAEM-*co*-MAA) cryogel takes place, as shown in Figure 12. The uptake of surfactants by P(DMAEM-*co*-MAA) cryogel is probably accounted for by both electrostatic and hydrophobic interactions of surfactant molecules.

Upon binding of CTMAC, the rate of water flux through the cryogel significantly decreases by 44% in comparison with the initial rate. Hydrophobization of pore surfaces of P(DMAEM-*co*-MAA) cryogel by CTMAC seems to retard the water flux through the pores. According to the conclusion of authors [36] the surfactant aggregates inside the cryogel matrix, irrespective of whether the attraction mechanism is electrostatic or hydrophobic.

There are two ways of immobilization of proteins within polyampholyte cryogels. The first approach is adsorption of proteins by preliminary prepared cryogels and the second one is imprinting of proteins within a cryogel network in situ cryopolymerization conditions. 

It is well known that the binding of proteins by cryogels has an electrostatic nature, and takes place between their isoionic point (IIP) and isoelectric point (IEP) [28]. Figure 13 shows the amount of adsorbed proteins by amphoteric cryogel after one day.

The maximal adsorption capacity (q_m_) of P(DMAEM-*co*-MAA) cryogel with respect to dyes, surfactants and proteins at different pHs is summarized in Table 4.

The binding ability of P(DMAEM-*co*-MAA) cryogel changes in the following order: SDBS > lysozyme >> MO > MB.

One of the unique fundamental property of polyampholytes is the so called isoelectric effect, which takes place at the IEP [28,37,38,39,40,41,42]. Therefore, the feasibility of utilizing such a phenomenon for polyampholyte cryogels was tested. The IEP of equimolar P(DMAEM-*co*-MAA) found from the flow rate experiments [31] was equal to 7.1 (Figure 14).

Table 5 shows the application of the isoelectric effect with respect to SDBS, MB, MO, and lysozyme [31]. 

At pH 5.3, the amount of MO and MB washed out from the cryogel volume is small, equal to 5.5% and 6.7%, respectively. At pH 9.5, the amount of lysozyme and MB released from the cryogel volume is also small, equal to 2% and 6% respectively. However, at the IEP of P(DMAEM-*co*-MAA) cryogel (pH 7.1), the amount of released MO, MB, SDBS and lysozyme released from the cryogel volume reaches up to 93–98% (Figure 15).

The release mechanism of low- and high molecular compounds at the IEP of P(DMAEM-*co*-MAA) is explained in Figure 16. 

At the IEP of P(DMAEM-*co*-MAA) and at pH of 7.1 the cooperativity of intrachain interactions between the acidic and basic groups (formation of inner salt) within a cryogel matrix predominates those of interchain interactions between cryogel and protein (surfactant and dye molecules). As a result, at the IEP of amphoteric cryogel (pH_IEP_ = 7.1) the lysozyme, MB, MO and SDBS are released from the cryogel matrix to the outer solution.

Molecularly imprinted polyampholyte (MIP) hydrogels and cryogels [43,44,45,46] are promising materials for biomedical applications [47,48,49]. Macroporous amphoteric polymers based on *N*-[3-(dimethylamino)propyl]methacrylate and MAA [46] and *N*,*N*-dimethylaminopropylacrylamide and allylamine (AA) [50] were used as a templates, adsorbents and carriers with respect to bovine serum albumin (BSA) [50] and lysozyme [46]. Macroporous interpenetrating amphoteric cryogels containing methacrylic acid (1^st^ network) and crosslinked chitosan (2^nd^ network) exhibit excellent adsorption-desorption properties with respect to lysozyme [51]. 

The adsorption capacity of MIP with respect to BSA is much higher (~95%) than nonimprinted polyampholyte (NIP) cryogels (~24%) but the mass transmission resistance is lower in cryogels compared with conventional nonmacroporous materials [45]. As seen in Figure 17, the adsorption saturation for NIP cryogel was reached within 20 min. In the case of MIP, saturation took 50 min because of fitting of template to recognizing sites.

For investigation of the imprinting capability of polyampholyte cryogels, the proteins lysozyme, pepsin, ovalbumin, hemoglobin, and γ-globulin were chosen as templates [52]. The molecular weights of proteins were in the range of (14–210) kDa, the IEPs of proteins varied from 1 to 11 while the IEP of P(AA-*co*-AAc-*co*-AAm) was estimated at about 4.5. An imprinting factor (IF) was calculated by the formula: *IF* = *k*_MIP_/*k*_NIP_ (where *k*_MIP_ and *k*_NIP_ are the retention factors on the molecularly imprinted polyampholyte—MIP and nonimprinted polyampholyte—NIP) resulting in an IF range of 2–7, which changed in the following order: lysozyme >> ovalbumin > hemoglobin > pepsin > γ-globulin. Thus, polyampholyte cryogels are suitable materials for separation and analysis of proteins and hopefully will play an important role in molecular recognition, separation science and biocatalysis.

Researchers [53] have prepared lysozyme imprinted polyampholyte hydrogel made of acrylamide, methacrylic acid and *N*,*N*-dimethylaminoethylmethacrylate. The amount of the lysozyme template able to be extracted reached up to 83%. The selective adsoprtion tests showed an IF equal to 3.38 for lysozyme.

Molecularly imprinted polyampholyte hydrogel with respect to BSA was prepared from acrylamide (AAm), *N*-isopropylacrylamide (NIPAM), [2-(methacryloyloxy)ethyl]trimethylammonium chloride (METMAC), and 2-acrylamido-2-methyl-1-propanesulfonic acid (AMPS) monomers with MBAA as the crosslinker [54]. The reason of selecting such monomers is to provide an opportunity for the protein the hydrogen bond, as the hydrophobic environment and electrostatic interaction increase the selctivity of the hydrogel matrix to recognize the key protein. The morphology of molecularly imprinted polyampholyte hydrogels (MIPAHs) is different from nonimprinted polyampholyte hydrogels (NIPAHs). The pore diameter of MIPAHs is larger than NIPAHs. In order to optimize the structural and functional affinity, the ideal conditions for preparing MIPAHs were found, including AAm concentration, NIPAM/AAm molar ratio, charge densitiy ratio (expressed as METMAC/AMPS molar ratio), and crosslinking density. The maximal binding of BSA (1.97 mg/g) and IF (2.91) was found for equimolar polyampholyte hydrogel. It is suggested that in the case of quenched polyampholyte gels, the specific rebinding and selectively recognition of imprinted proteins becomes tailorable. 

Polybetainic (or polyzwitterionic) hydrogels and cryogels based on poly{[2-(methacryloyloxy)ethyl]dimethyl-(3-sulfopropyl) ammonium hydroxide}, PMODMSPA with excellent biocompatibility and antofouling properties were prepared by both conventional polymerization at room temperature and under cryopolymerization conditions [27,55]. Polyampholyte cryogels showed high protein loading efficiency at up to 80% and a four-month sustained release rate of a model protein—BSA with very little burst release, whereas the conventional polyampholyte hydrogel showed lower protein loading efficiency (~45%) and a high burst release. The release of BSA from PMODMSPA hydrogel, poly(2-hydroxyethylmethacrylate) (polyHEMA) hydrogel and cryogel matrix continued 4 weeks, whereas the maximal release of BSA from PMODMSPA cryogel reached up to 4 months. 

Thus, the adsorption capacity of molecularly imprinted polyampholyte cryogels with respect to proteins is much higher than nonimprinted polyampholyte cryogels. However, the adsorption saturation time for MIP cryogel is longer than nonimprinted polyampholyte cryogel because of fitting of the template to recognizing sites. Zwitterionic cryogels will prove useful for prolonged macromolecular therapeutic delivery and tissue engineering applications.

## 4. Immobilization of Metal Nanoparticles within Polyampholyte Cryogel Matrix and Evaluation of Their Catalytic Activity

Cryogel catalysts are promising materials due to the inherent features: macroporous structures, adjustable hydrophilicity and hydrophobility, stimuli-sensitivity and combination of catalytic groups [55,56,57]. Both polyampholyte cryogel itself [18] and cryogel-immobilized metal nanoparticles [58,59,60,61,62,63,64] exhibit unique catalytic properties for various chemical reactions. Nonionic [59,60], anionic [61,62], and cationic [63,64] supermacroporous cryogels and their templated metal nanoparticle composites (Co, Ni, Cu, and Fe) have been used in the redox reaction of NaBH_4_ and hydrogenation of nitrogroup containing substrates, such as 4-nitrophenol (4-NP), 2-nitrophenol (2-NP) and 4-nitroaniline (4-NA). 

Modified polyacrylamide-based amphoteric cryogels were tested as catalytically active substances in transesterification of glyceryl oleate [18]. The highest conversion of glyceride was equal to 88.4% and the yield of methyl oleate is about 64.%. 

The catalytic reduction of 4-NP by NaBH_4_ and oxidation of d,l-dithiotreitol (DTT) by hydrogen peroxide in the presence of AuNPs supported on a cryogel matrix of P(DMAEM-*co*-MAA) was evaluated, and the kinetic parameters, turnover number (TON), turnover frequency (TOF) and activation energy of hydrogenation of 4-NP have been calculated [65,66,67,68]. 

Porous P(DMAEM-*co*-MAA) cryogels with immobilized AuNPs were used as a flow-through catalytic reactor in the reduction of 4-NP and oxidation of DTT. The final hydrogenation product of 4-NP is 4-aminophenol (4-AP), while the final oxidation product of DTT is disulfide (DS) (Figure 18).

Both hydrogenation of 4-NP and oxidation of DTT was carried out in a flow-through catalytic reactor, represented in Figure 19.

Passing of the mixture of 4-NP and NaBH_4_ through P(DMAEM-*co*-MAA) cryogel containing AuNPs leads to reduction of 4-NP to 4-AP [65]. The activation energy of 4-NP reduction in the presence of P(DMAEM-*co*-MAA)/AuNPs is equal to 7.52 kJ·mol^−1^, which is 2–5 times lower compared to data presented in previous literature [61,62,64]. The AuNPs immobilized cryogel catalyst sustained over 50 cycles without substantial loss of the catalytic activity. The size and shape of AuNPs immobilized within the cryogel matrix are considerably changed after 50 cyclic reductions of 4-NP in comparison with the initial state [65]. The average size of AuNPs accumulated in both surface and longitudinal parts of the cryogel was less than 100 nm. Moreover, the partial leaching out of AuNPs was observed over the course of reduction of sequental portions of substrate [68]. In reduction of 4-NP, the values of TON and TOF of the cryogel catalyst are equal to 38.17 and 21.56 h^−1^ respectively. The cryogel catalyst P(DMAEM-*co*-MAA)/AuNPs also shows the high catalytic activity in oxidation of DTT and sustains 10 cyclic oxidation of substrate with 97–98% conversion. The values of TON and TOF after 10 cyclic oxidation of DTT are equal to 985.2 and 412.2 h^−1^ respectively. The oxidation of DTT proceeds more effectively than reduction of 4-NP due to a high rate constant.

Thus cryogel catalysts based on P(DMAEM-*co*-MAA)/AuNPs exhibit a high catalytic activity and conversion in reduction of 4-NP and oxidation of DTT; they can serve as a flow-through catalytic reactor and provide a cascade-type reaction without isolation of intermediate products. The main advantage of the macroporous structure is a large contact area of catalyst and substrate, reusability and simplicity of separation of products from the reaction medium.

The catalytic reduction of *p*-nitrobenzoic acid (*p*-NBA) to *p*-aminobenzoic acid (*p*-ABA) was performed using palladium (PdNPs) and gold nanoparticles (AuNPs) supported on a cryogel matrix of P(DMAEM-*co*-MAA) [69]. 

Transformation of *p*-NBA to *p*-ABA over DMAEM-*co*-MAA/AuNPs catalyst proceeds via formation of intermediate compounds [70] (Figure 20). 

It should be noted that in the absence of immobilized AuNPs the mixture of *p*-NBA and NaBH_4_ fluxed through the P(DMAEM-*co*-MAA) cryogel does not produce *p*-ABA. In the course of *p*-NBA hydrogenation except of *p*-ABA the formation of two additional products was detected. One of them is *p*,*p*’-azodibenzoate, which is the product of catalytic coupling condensation of nitroso compound with hydroxylamine (Figure 21).

The proposed mechanism of formation of *p*,*p*’-azodibenzoate on P(DMAEM-*co*-MAA)/AuNPs is confirmed by Raman spectroscopy [71] and is in good agreement with photoreduction of *p*-NBA on nanostructured silver through photoinduced surface catalytic coupling reactions [72]. 

Another product of *p*-NBA conversion is sodium 4-(4-aminobenzamido)benzoate formed as a result of condensation of amine and carboxylate groups, as demonstrated in Figure 22. A direct condensation polymerization of N-alkylated *p*-ABA has been performed by researcher [73].

Thus, the hydrogenation of *p*-NBA over DMAEM-*co*-MAA/AuNPs produces at least three products: amino-, azo- and amido-derivatives.

In case of P(DMAEM-*co*-MAA)/PdNPs the formation of by-products is not observed. As seen in Table 6, approximately 40% of *p*-NBA is converted to *p*-ABA at [*p*-NBA]:[NaBH_4_] = 1:50 mol/mol, whereas the conversion is 100% at [*p*-NBA]:[NaBH_4_] = 1:200 mol/mol. 

The rate constants and activation energy of *p*-NBA reduction over DMAEM-*co*-MAA/PdNPs are represented in Figure 23. The E_a_ of *p*-NBA reduction on DMAEM-*co*-MAA/PdNPs is equal to 38.83 kJ·mol^−1^.

Thus, amphoteric cryogels with immobilized palladium and gold nanoparticles may serve as effective flow-through units for continuous hydrogenation or oxidation of various substrates and provide a new strategy for the syntheses of amino compounds, azo dyes, or benzamides from aromatic nitro compounds.

## 5. Comparative Analysis of Polyampholyte Cryogels with Nonionic, Anionic and Cationic Precursors

Table 1 of the review [3] summarizes the preparation protocol, morphology and physicochemical properties of nonionic cryogels based on AAm, *N*,*N*-dimethyl (or diethyl) acrylamide (DMAAm, DEAAm), NIPAM, 2-hydroxyethylmethacrylate (HEMA); anionic cryogels synthesized using homo- or copolymerization of AAc, MAA, AAm with AMPS, AAm with AAc or itaconic acid (IAc); and cationic cryogels prepared from DMAAm and 2-(diethylamino)ethylmethacrylate (DEAEM). Mostly, the thermo- or pH-sensitive behavior of the above mentioned cryogels were studied. Examples of nonionic, anionic and cationic cryogels prepared from synthetic or natural macromolecular precursors such as poly(vinyl alcohol) (PVA), poly(ethylene oxide) (PEO), poly(*N*-vinylpyrrolidone) (PVP), chitosan, pectin, and cellulose, among others are summarized in Table 2 [3].

In spite of the fact that cryogels of proteins—ovalbumin, serum albumin, gelatin described in Table 3 of [3]—belong to polyampholyte cryogels too, their amphoteric character is weakly exhibited due to a tiny percentage of aminoacids in the macromolecular chain. In contrast, the synthetic polyampholyte cryogels are specific multifunctional objects and have excellent acid–base or anionic–cationic character as well as demonstrate adequate swelling-deswelling, and collapsing behavior in response to external factors such as temperature, pH, ionic strength, metal ions, the nature and charge of low-molecular-weight anions and cations, the mixture of water–organic solvent, and so forth. Some fundamental properties discovered for water-soluble polyampholytes, such as the antipolyelectrolyte effect and the isoelectric effect [37] were clearly demonstarted in the case of polyampholyte cryogels. The antipolyelectrolyte effect that is related to the unfolding of amphoteric macromolecules at the isoelectric point (IEP) upon addition of low-molecular-weight salts was shown for polyampholyte cryogels (see Figure 6). The isoelectric effect that is related to cooperative release of low- or high molecular-weight substances (such as metal ions, dyes, surfactants, polyelectrolytes and proteins) at the IEP of water-soluble polyampholytes also takes place at the IEP of polyampholyte cryogels (see Table 5 and Figure 15). Both phenomena can have practical applications in terms of water desalination, removal of metal ions and organic ions from the wastewater as well as for separation and purification of proteins, and as drug delivery systems. 

In future, the closeness of polyampholyte cryogels to cryogels of polyelectrolyte complexes—the products of the interactions between oppositely charged polyelectrolytes might be demonstrated. Polyampholyte cryogels are able to form a very stable metal chelates within macropores due to formation of both ionic and coordination bonds with acidic and basic groups, and are able to perform specific sorption of metal ions according to stability constants. Recovery of transition metal ions and organic impurities by polyampholyte cryogels is especially important for hydrometallurgy processes and environmental protection. In its turn, the reduction of polyampholyte cryogel-transition metal complexes by reducing agents can produce the nano- and micron-sized particles of metals and/or metal oxides on the inner and surface parts of amphoteric cryogels that might be used as efficient flowing microreactors in catalysis. Passing the mixture of substrate and reducing agent through the cryogel pores provides enough contact time between the catalyst and reaction mixture. 

The amphoteric cryogels may retain the amphoteric substrate molecules on the surface of the cryogel matrix, prolonging the contact time between substrate and catalyst. For instance, amphoteric cryogel matrix (DMAEM-MAA) retains the amphoteric product of hydrogenation (p-ABA) much longer and provides more effective catalytic conversion of a substrate. This is the reason of further conversion of amine groups of p-ABA to azocompound and benzamide (see Figure 21 and Figure 22). One can expect that in near future the polyampholyte cryogels may be an effective tool for synthesis of polypeptides from aminoacids in appropriate catalytic conditions. The microstructure of polyampholyte cryogels can be expanded to a random, alternating, graft, block, or dendritic structure with combinations of weak acid/weak base, strong acid/weak base or else weak acid/strong base, and strong acid/strong base.

## 6. Conclusions

Polyampholyte cryogels as distinct from nonionic, anionic and cationic precursors are unique materials due to the simultaneous presence of acid–base or anionic–cationic monomer units in the macromolecular chain, which considerably expand our opportunities and knowledge about this subject. The behavior of polyampholyte cryogels is not much deviated from the properties of most linear polyampholytes and polyampholyte hydrogels. At the same time, polyampholyte cryogels have many advantages related to applicability. They can be used for removal of metal ions, dyes, surfactants, and drugs from the wastewater. Binding, separation and purification of target proteins from a mixture can also be performed with the help of molecularly imprinted polyampholyte cryogels. Embedding of metal ions or metal nanoparticles within polyampholyte cryogels provides a chance to design active, stable, selective, easy to handle and reusable catalytic systems. To our knowledge the concept of “green chemistry” in the context of catalytic chemistry means reaction behavior in mild conditions, e.g., at atmospheric pressure and low temperature including the easy separation of products from catalysts. These requirements can successfully be realized in case of flow through catalytic reactors made of macroporous polymers within of which metal nanoparticles or enzymes are immobilized. An open problem in the behavior of polyampholyte cryogels is the use of the antipolyelectrolyte effect, i.e., the ability of functional groups of amphoteric cryogels to accumulate low-molecular-weight salts at the IEP, such as for desalination of saline water. Another opportunity is the realization of the “isoelectric” effect for purification and separation of cells, DNA, nucleotides, and proteins from a mixture. Furthermore, theoretical, computational and simulation approaches for better understanding the physicochemical, complexation and catalytic properties of polyampholyte cryogels are required.

## Figures and Tables

**Figure 1 gels-05-00008-f001:**
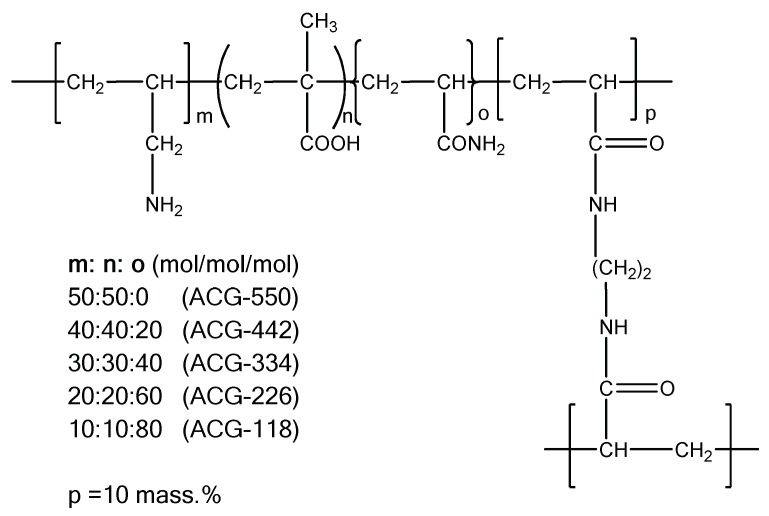
Structure and composition amphoteric cryogel (ACG) cryogels derived from AA, MAA and AAm. (Reprinted with permission from [29], copyright 2012 Wiley).

**Figure 2 gels-05-00008-f002:**
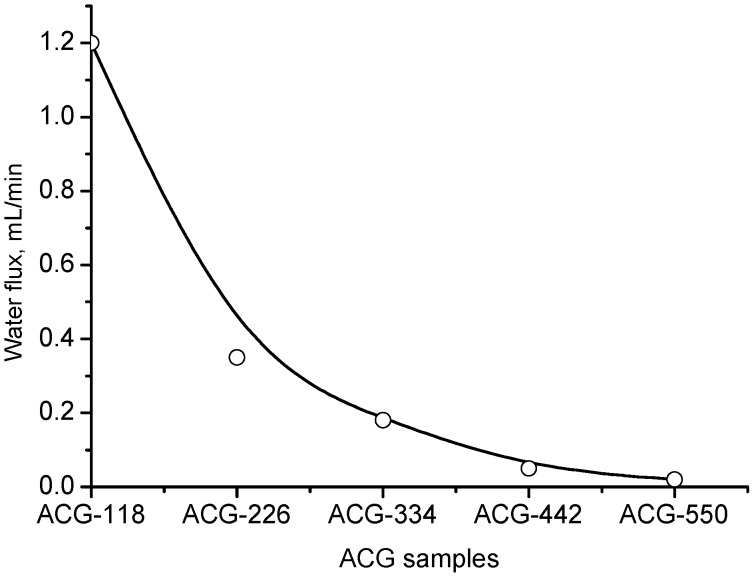
Dynamics of water flux through the ACG samples. (Adapted with permission from [29], copyright 2012 Wiley).

**Figure 3 gels-05-00008-f003:**
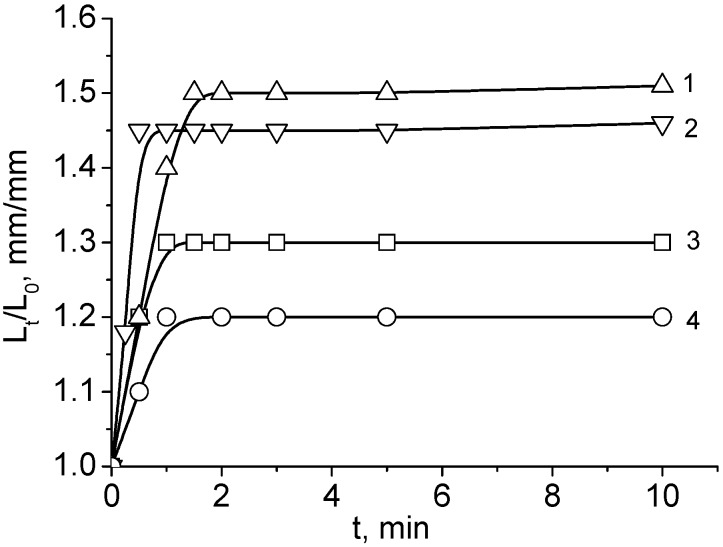
Time dependent swelling of ACG-118 (**1**), ACG-226 (**2**), ACG-334 (**3**) and ACG-550 (**4**) in water. For abbreviation of cryogels please see Figure 1. (Reprinted with permission from [29], copyright 2012 Wiley).

**Figure 4 gels-05-00008-f004:**
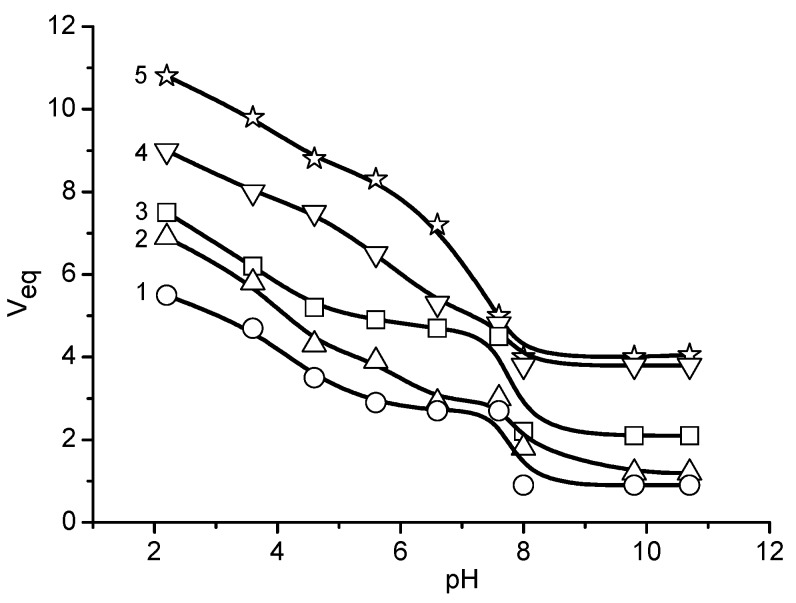
The pH dependent equilibrium swelling (V_eq_) of P(DMAEM-*co*-AMPS) cryogels. The mole fraction of AMPS is 0 (**1**), 5 (**2**) 15 (**3**), 25 (**4**) and 40 mol.% (**5**). (Updated from [24]).

**Figure 5 gels-05-00008-f005:**
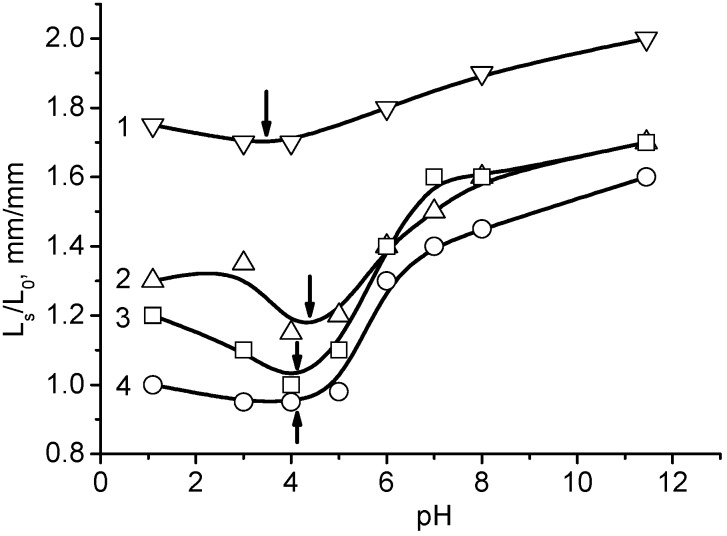
pH-dependent swelling-deswelling of ACG-118 (**1**), ACG-226 (**2**), ACG-334 (**3**) and ACG-550 (**4**). Arrows indicate on the position of the pH_IEP_. (Reprinted with permission from [29], copyright 2012 Wiley).

**Figure 6 gels-05-00008-f006:**
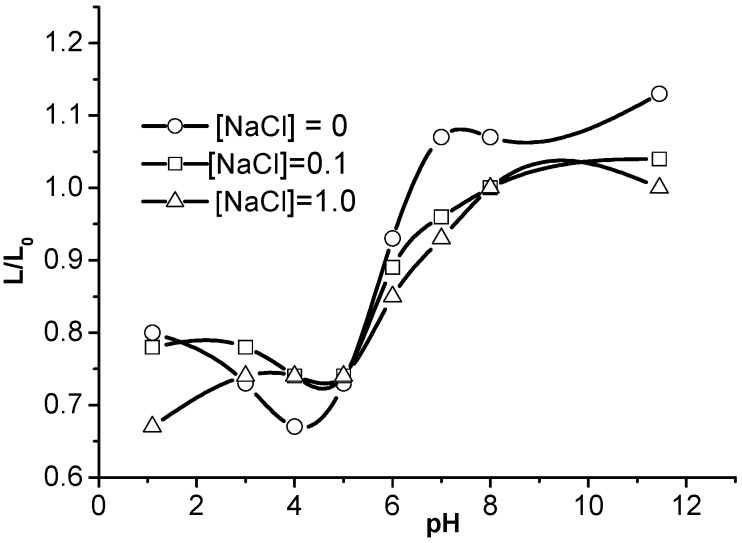
pH-dependent swelling of ACG-334 at various ionic strengths. (Reprinted with permission from [29], copyright 2012 Wiley).

**Figure 7 gels-05-00008-f007:**
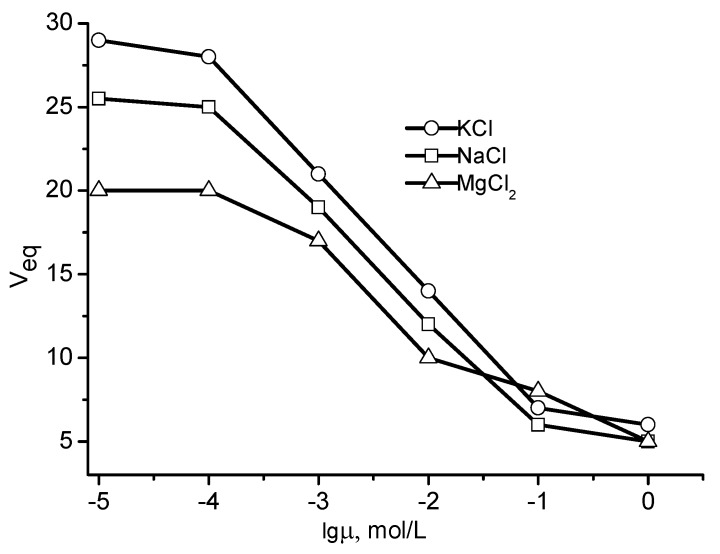
Swelling curves of P(DMAEM-*co*-AMPS) cryogels in aqueous solutions of NaCl, KCl, MgCl_2_. [AMPS] = 40 mol.%. (Redrawn from [24]).

**Figure 8 gels-05-00008-f008:**
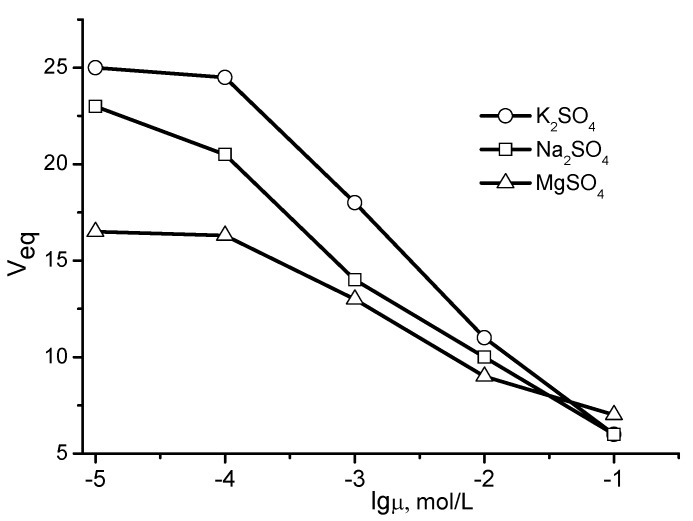
Swelling curves of P(DMAEM-*co*-AMPS) cryogels in aqueous solutions of K_2_SO_4_, Na_2_SO_4_, and MgSO_4_. [AMPS] = 40 mol.%. (Redrawn from [24]).

**Figure 9 gels-05-00008-f009:**
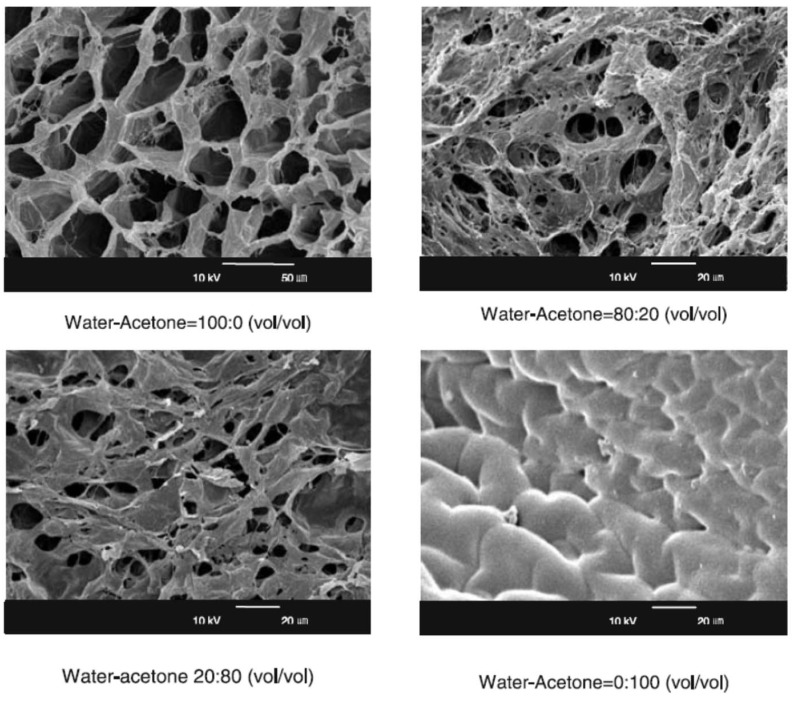
Scanning electron microscope (SEM) pictures of polybetainic gel in water, acetone and water–acetone mixture (Reprinted with permission from [30], copyright 2006 Springer).

**Figure 10 gels-05-00008-f010:**
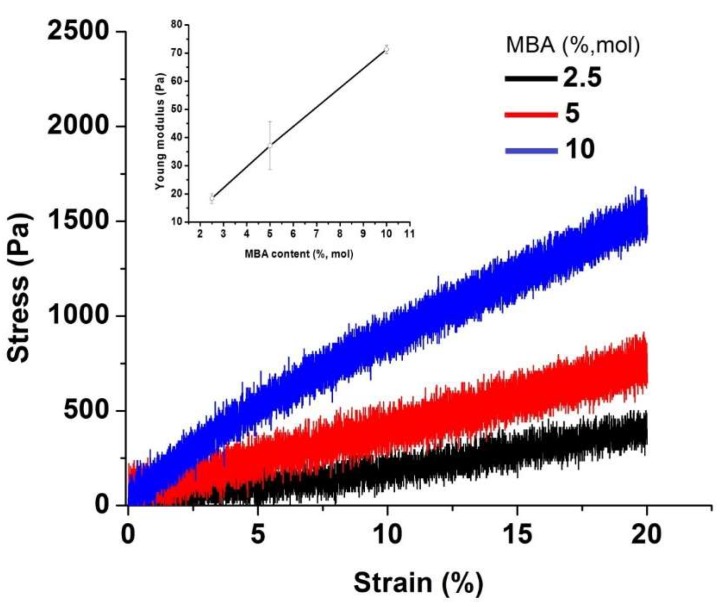
Mechanical properties of P(DMAEM-*co*-MAA) cryogels at different concentrations of MBAA. (Reprinted with permission from [31], copyright 2016 Wiley).

**Figure 11 gels-05-00008-f011:**
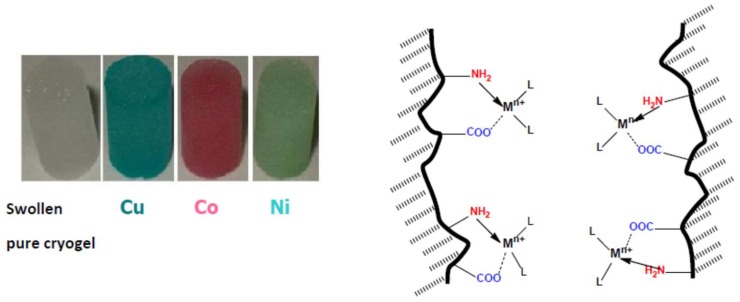
Complexes of ACG-334 with Cu^2+^, Ni^2+^ and Co^2+^ ions (**left**) and polymer-metal complexation in cryogel pores (**right**). (Reprinted with permission from [29], copyright 2012 Wiley).

**Figure 12 gels-05-00008-f012:**
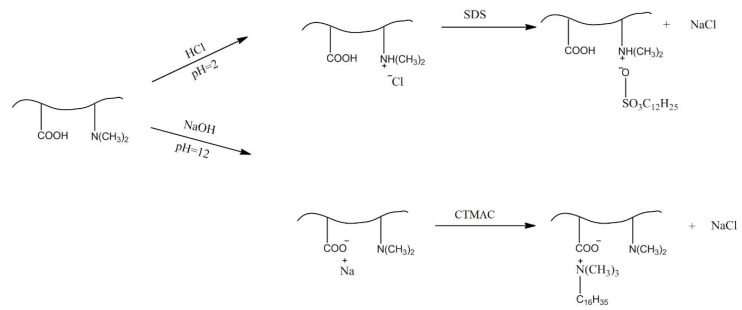
Interaction of SDBS (**upper route**) and CTMAC (**lower route**) with amphoteric cryogel P(DMAEM-*co*-MAA).

**Figure 13 gels-05-00008-f013:**
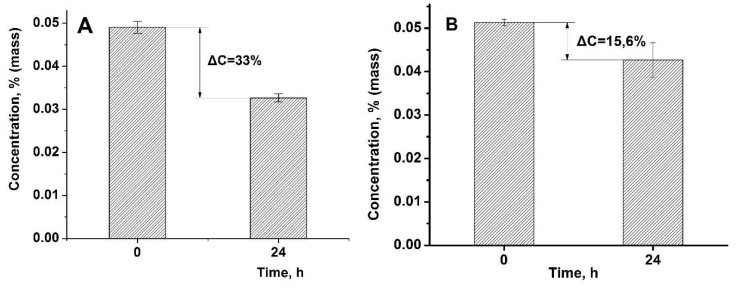
Changing of lysozyme (**A**) and cytochrome C (**B**) concentrations after one day contacting with P(DMAEM-*co*-MAA) cryogel.

**Figure 14 gels-05-00008-f014:**
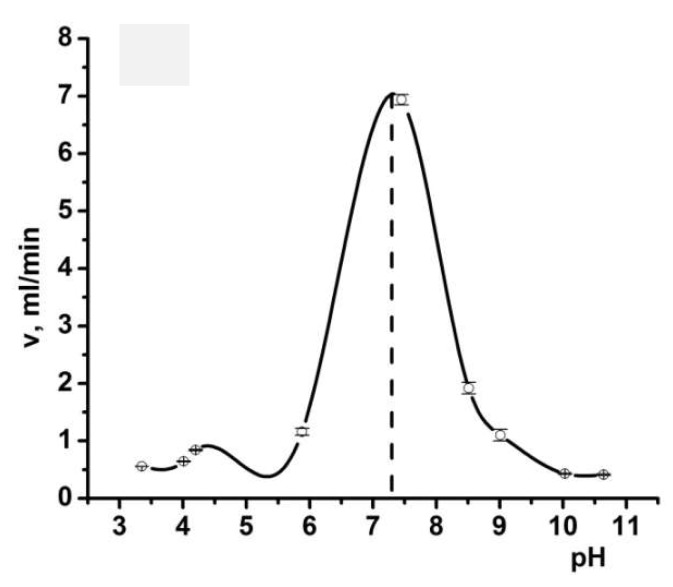
pH dependent change in the flow rate through equimolar P(DMAEM-*co*-MAA) cryogels. The dotted line indicates on the position of the isoelectric point (IEP). (Reprinted with permission from [31], copyright 2016 Wiley).

**Figure 15 gels-05-00008-f015:**
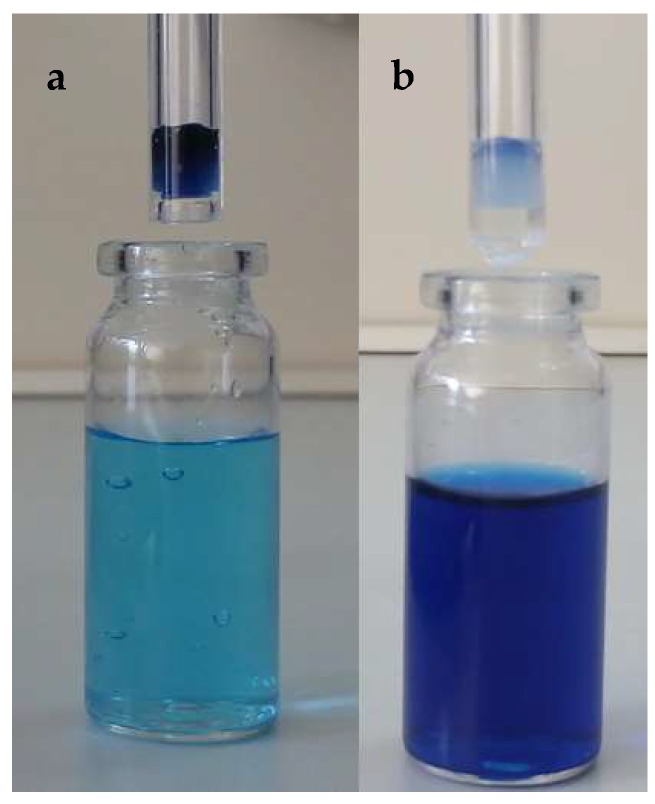
Cryogel samples washed out by buffer solution at pH 9.5 (**a**) and at the IEP pH 7.1 (**b**). (Reprinted with permission from [31], copyright 2016 Wiley).

**Figure 16 gels-05-00008-f016:**
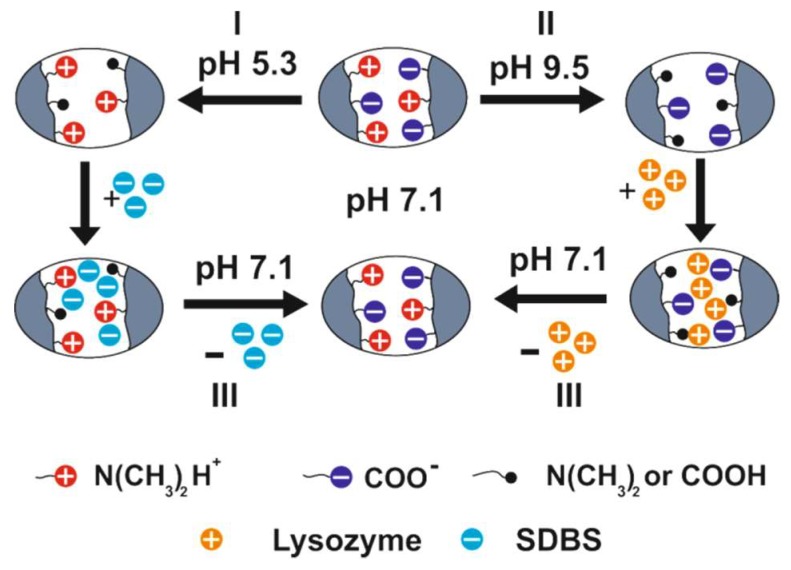
Schematic representation of adsorption of SDBS at pH 5.3 (**route I**), lysozyme at pH 9.5 (**route II**) and their release at pH 7.1, e.g., at the IEP of P(DMAEM-*co*-MAA) (**route III**). (Reprinted with permission from [31], copyright 2016 Wiley).

**Figure 17 gels-05-00008-f017:**
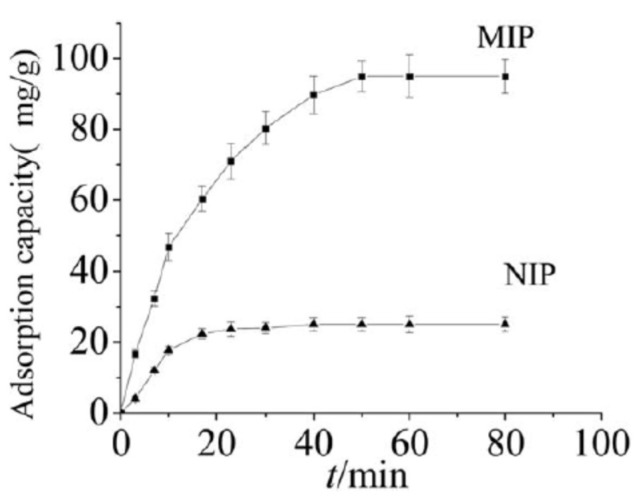
Binding kinetics of bovine serum albumin (BSA) by molecularly imprinted polyampholyte (MIP) and nonimprinted polyampholyte (NIP) based on P(AA-*co*-AAc-co-AAm) cryogels. (Reprinted with permission from [21], copyright 2016 Wiley).

**Figure 18 gels-05-00008-f018:**
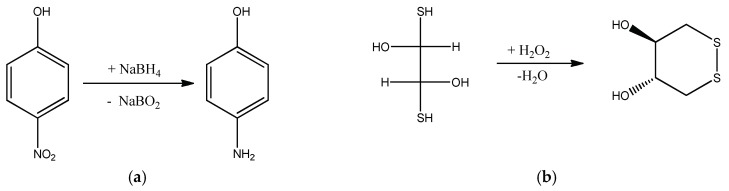
Schemes of reduction of 4-NP by NaBH_4_ (**a**) and oxidation of d,l-dithiotreitol (DTT) by hydrogen peroxide (**b**).

**Figure 19 gels-05-00008-f019:**
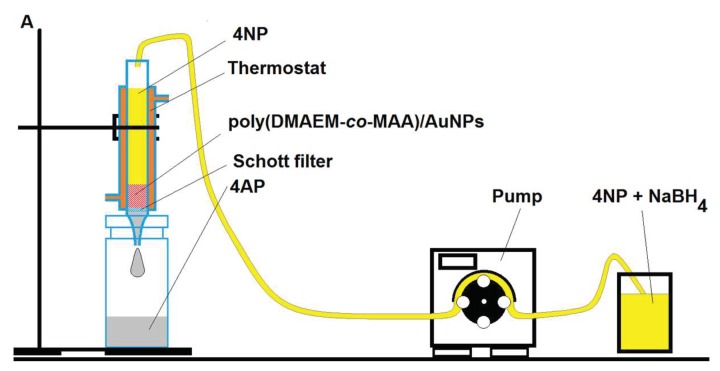
Schematic representation of 4-NP reduction and DTT oxidation by flow-through catalytic reactor based on P(DMAEM-*co*-MAA)/AuNPs. (Reprinted with permission from [67], copyright 2016 Wiley).

**Figure 20 gels-05-00008-f020:**
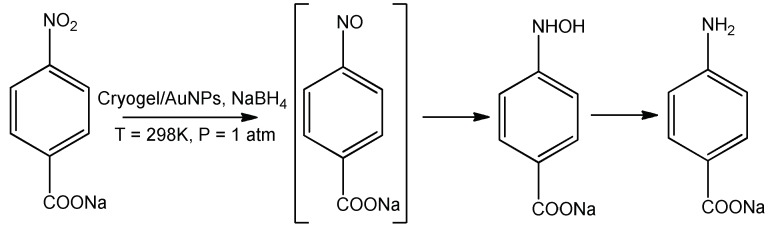
Hydrogenation of *p*-NBA over P(DMAEM-*co*-MAA)/AuNPs catalyst.

**Figure 21 gels-05-00008-f021:**
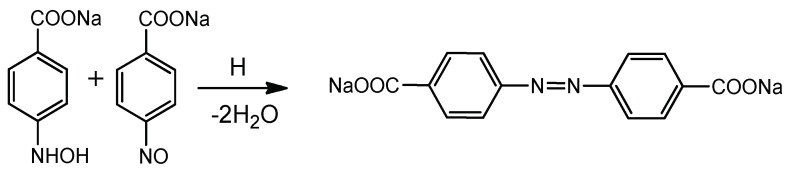
The proposed mechanism of formation of *p*,*p*’-azodibenzoate.

**Figure 22 gels-05-00008-f022:**
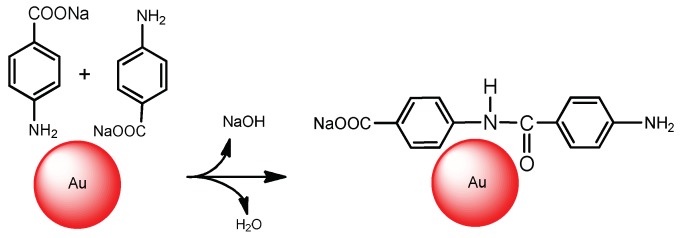
The proposed mechanism of formation of sodium 4-(4-aminobenzamido)benzoate.

**Figure 23 gels-05-00008-f023:**
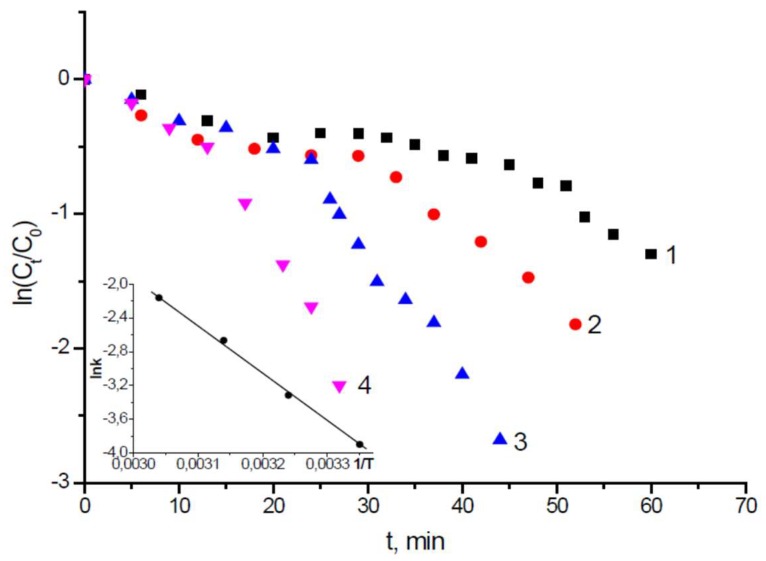
Kinetic curves of *p*-NBA reduction over P(DMAEM-*co*-MAA)/PdNPs are expressed in the graph ln(C_t_/C_o_) *vs* time. The insert shows lnk *vs* 1/T. T = 298 (**1**), 308 (**2**), 318 (**3**) and 328 K (**4**). (Reprinted from [69], copyright 2018 Springer).

**Table 1 gels-05-00008-t001:** Examples of various polyampholyte cryogels.

Chemical Structure of Polyampholyte Cryogels *	Name Acronym	Refs
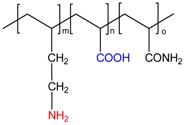	allylamine-*co*-acrylic acid-*co*-acrylamide, P(AA-*co*-AAc-*co*-AAm)	[18]
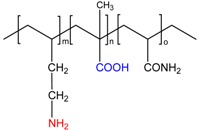	allylamine-*co*-methacrylic acid-*co*-acrylamide, P(AA-*co*-MAA-*co*-AAm)	[19,20]
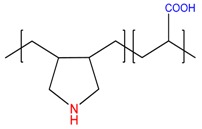	diallylamine-*co*-acrylic acid, P(DAA-*co*-AAc)	[21]
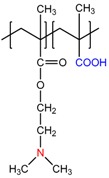	*N*,*N*-dimethylaminoethylmethacrylate-*co*-methacrylic acid, P(DMAEM-*co*-MAA)	[22,23]
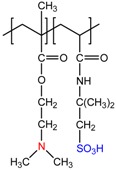	*N*,*N*-dimethylaminoethylmethacrylate-*co*-2-acrylamido-2-methyl-1-propanesulfonic acid, P(DMAEM-*co*-AMPS)	[24]
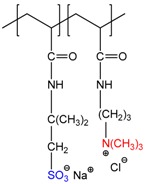	2-acrylamido-2-methyl-1-propanesulfonic acid sodium salt-*co*-(3-acrylamidopropyl)trimethylammonium chloride, P(AMPSNa-*co*-APTAC)	[25,26]
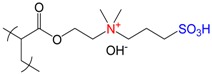	Poly{[2-(methacryloyloxy)ethyl]dimethyl-(3-sulfopropyl)ammonium hydroxide}, PMODMSPA	[27]

* Primary, secondary, tertiary amine and quaternary ammonium groups of polyampholytes are in red, carboxylic and sulfonic groups are in blue.

**Table 2 gels-05-00008-t002:** Acid–base content, pK_b_ of allylamine groups and the isoelectric pH (pH_IEP_) of ACG cryogels. (Reprinted with permission from [29], copyright 2012 Wiley).

Cryogels	–NH_2_, mol.%		–COOH, mol.%		pK_b_	pH_IEP_
	Potentiometric Titration	Conductimetric Titration	Potentiometric Titration	Conductimetric Titration		
ACG-550	46.7	41.0	53.3	59.0	5.44	4.0
ACG-442	42.4	37.0	57.6	63.0	5.25	4.1
ACG-334	43.3	33.3	56.7	66.6	5.78	4.2
ACG-226	-	41.4	-	58.6	5.62	4.3
ACG-118	-	38.5	-	61.5	-	3.5

**Table 3 gels-05-00008-t003:** Adsorbed and desorbed amounts of metal ions by ACG-334. (Reprinted with permission from [29], copyright 2012 Wiley).

Transition Metal Ions	Concentration of Metal Ions, mol·L^−1^	Adsorbed, %	Desorbed, %
Cu^2+^	10^−3^	99.9	51.4
Ni^2+^		99.9	67.2
Co^2+^		99.9	62.0

**Table 4 gels-05-00008-t004:** Adsorption of methylene blue (MB), methyl orange (MO), sodium dodecylbenzenesulfonate (SDBS), and lysozyme by P(DMAEM-*co*-MAA) cryogel. (Reprinted with permission from [31], copyright 2016 Wiley).

Substances	q_m_, mg·g^−1^	pH of Adsorption
Lysozyme	123	9.5
MB	8.3	9.5
SDBS	1313	5.3
MO	27.1	5.3

**Table 5 gels-05-00008-t005:** Detachment of MB, MO, SDBS and lysozyme from P(DMAEM-*co*-MAA) cryogel matrix at different pH. (Reprinted with permission from [31], copyright 2016 Wiley).

Substance	Release, mg/%		
	pH		
	7.1 (IEP)	9.5	5.3
MB	0.04/94	0.003/6	-
MO	0.13/94.5	-	0.0076/5.5
Lysozyme	1.72/98	0.035/2	-
SDBS	17.9/93.2	-	1.3/6.7

**Table 6 gels-05-00008-t006:** Conversion degree of *p*-NBA to *p*-ABA over DMAEM-*co*-MAA/PdNPs at various molar ratios of substrate to reducing agent. (Reprinted from [69], copyright 2018 Springer).

Cycles	Conversion Degree, %		
	[*p*-NBA]:[NaBH_4_] = 1:50 mol/mol	[*p*-NBA]:[NaBH_4_] = 1:100 mol/mol	[*p*-NBA]:[NaBH_4_] = 1:200 mol/mol
1	0.95	100	100
2	39.02	89.84	100
3	34.73	82.77	100
4	38.10	78.48	100
5	38.40	78.78	100
6	39.63	78.78	100
7	39.94	80.01	100
8	39.94	84.34	100
9	40.55	84.92	100
10	44.23	84.62	100
Average	39.93 *	84.28	100

* Excepting for the first cycle.

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
