# Peer review of "Physicochemical, Complexation and Catalytic Properties of Polyampholyte Cryogels"

_gels, 2019, doi:10.3390/gels5010008_

Round 1
Reviewer 1 Report
Overall this manuscript provides a thorough review of polyampholyte cryogels. There are a few minor editorial concerns that must be addressed prior to publication, but these can be easily fixed.
In Figure 3, two of lines lines are labeled as species 1 and no line is labeled as species 4.
In Figure 4, none of the lines are labeled with the corresponding species number from the caption.
In Figure 6, again two of the lines are labeled as species 1 and no line is labeled as species 4.
Once these minor concerns are corrected, I believe this manuscript would be of interest to the readers of Gels.
Author Response
Response to Reviewer 1 Comments
Point 1: In Figure 3, two of lines are labeled as species 1 and no line is labeled as species 4.
Response 1: In Figure 3 the last line 1 is changed to line 4.
Point 2: In Figure 4, none of the lines are labeled with the corresponding species number from the caption.
Response 2: Figure 4 was redrawn and all lines were labeled accordingly
Point 3: In Figure 6, again two of the lines are labeled as species 1 and no line is labeled as species 4.
Response 3: In Figure 6 the last line 1 is corrected to line 4
Reviewer 2 Report
Title: Physico-Chemical, Complexation and Catalytic Properties of Polyampholyte Cryogels
Author: Sarkyt Kudaibergenov
Manuscript ID: gels-436708
The author is known for his work in the field of polymeric gels, with amphoteric properties. He has summarized the recent findings of polyampholyte gels, where the properties and the complexation of these materials were discussed. The review is a kind of a continued work of the previously published review about intra- and interpolyelecrolyte complexes of polyampholytes published at the end of 2018 (Ref 37). The structure of the work is adequate, after the properties section the complexation were discussed, followed by the evaluation of the catalytic activiies of the immobilized nanoparticles.
The work is a nice contribution to the discussed field, but is lacking of the comparison with gels without polyamphoteric properties, which could also highlight the importance of these materials as well as give a bit more information about the favorable properties of the discussed materials.
The review work is recommended for publication in Gels after minor revision, by adding some more information about the favorable properties of these materials compared to the similar gels without amphoteric properties.
Author Response
Response to Reviewer 2 Comments
Point 1: The work is a nice contribution to the discussed field, but is lacking of the comparison with gels without polyamphoteric properties, which could also highlight the importance of these materials as well as give a bit more information about the favorable properties of the discussed materials.
Response 1: Additional Section 5 entitled “Comparative analysis of polyampholyte cryogels with nonionic, anionic and cationic precursors” was added to highlight the importance of polyampholyte cryogels and to show their favorable properties in comparison with cryogels without polyampholytic behavior
Point 2: The review work is recommended for publication in Gels after minor revision, by adding some more information about the favorable properties of these materials compared to the similar gels without amphoteric properties.
Response 2: Brief information about the favorable properties of polyampholyte cryogels is presented in Section 5
This manuscript is a resubmission of an earlier submission. The following is a list of the peer review reports and author responses from that submission.